# Polyurethane Foam Composites Reinforced with Renewable Fillers for Cryogenic Insulation

**DOI:** 10.3390/polym13234089

**Published:** 2021-11-24

**Authors:** Beatrise Sture, Laima Vevere, Mikelis Kirpluks, Daniela Godina, Anda Fridrihsone, Ugis Cabulis

**Affiliations:** Polymer Laboratory, Latvian State Institute of Wood Chemistry, Dzerbenes Street 27, LV-1006 Riga, Latvia; laima.vevere@kki.lv (L.V.); mikelis.kirpluks@kki.lv (M.K.); daniela.godina@kki.lv (D.G.); anda.fridrihsone@kki.lv (A.F.); ugis.cabulis@kki.lv (U.C.)

**Keywords:** rigid polyurethane foams, nanocellulose, silanization, cryogenic insulation

## Abstract

Sawdust, microcellulose and nanocellulose and their silanized forms were used to reinforce rigid polyurethane (PU) foam composites. The concentration of fillers was varied in the range of 0.5–1.5%. For rigid PU foam formulations, three polyols from recycled and renewable materials were used, among other components. Polyols were obtained from rapeseed oil, tall oil fatty acids and recycled polyethylene terephthalate. As rigid PU foam composites in literature have been described as appropriate thermal insulation material, the appliance of obtained composites for cryogenic insulation was investigated by determining the various physical-mechanical properties of composites. The physical-mechanical properties, such as the modulus of elasticity, compressive and tensile strength in both 293 K and 77 K, adhesion measurements with and without cryo-shock, apparent density, thermal conductivity coefficient, and safety coefficient were measured. The results showed that the addition of fillers did not give a significant improvement of characteristics.

## 1. Introduction

In recent years, the use of renewable and biodegradable raw materials has become the main research direction in the chemical industry, especially in the polymer field. Due to their excellent thermal insulation and mechanical characteristics, rigid polyurethane (PU) foams are one of the most commonly used type of PU materials [1,2], primarily in refrigerators, freezers, and other civil engineering applications [3]. Polyols from plant oils, such as rapeseed, soybean, sunflower, tall oil, have become a favorable alternative to the petrochemical-based polyols used in rigid PU foam manufacture [2,3,4].

Plant oils mainly consist of different fatty acids’ triglycerides. Since the double bonds in fatty acids are quite inert, the functionalization, e.g., epoxidation of double bonds, is required [3]. Most plant oil-based polyols are obtained by oxirane ring-opening with nucleophilic reagents [1,3].

Suitable polyols for rigid PU development can also be obtained from tall oil fatty acids (TOFA) (mixture of oleic and linoleic acid). Since tall oil is a by-product of cellulose pulping [5], it does not compete with the food and feed industry; therefore, it is a second-generation feedstock. Double bonds in oleic and linoleic acids in tall oil must be functionalized to efficiently obtain various polyols. Epoxidized tall oil fatty acids (ETOFA) and the subsequent oxirane ring-opening with triethanolamine (TEOA) have led to polyol with a high hydroxyl (–OH) value, which is suitable for rigid PU production [1,4].

Polyethylene terephthalate (PET) is one of the most versatile polymer materials, and it is commonly used as a packaging material, resulting in increased waste products [4]. Different recycling methods of PET have been investigated [6,7,8]. One of the chemical methods involves PET depolymerization in the presence of alcohols, acids, glycols, and amines. PET depolymerization yields aromatic polyester polyols, which can be used in the manufacturing of PU. For example, PET transesterification with diethylene glycol (DEG) yields polyester polyol [5,9]. Different by-products are obtained during PET glycolysis, which can decrease the functionality of polyols [10]. However, rigid PU foams from recycled PET polyols have lower shrinkage and density and better dimension stability [8,10].

Polymer composites reinforced with different natural fibers have been obtained for a long time [3,11,12,13]. As cellulose is the most easily approachable renewable polymer, it is being versatility used for obtaining bio-composites, mostly composites structured with nanosized cellulose [14,15]. The main reasons for using nanocellulose is its large distinctive surface area, low density, high strength and Young’s modulus, and low coefficient of thermal conductivity [15,16,17]. Nanofibrillated cellulose is used more commonly than nanocrystalline cellulose, as the obtained composites have higher tensile strength and elasticity modulus [18,19]. 

Natural fibers present a number of advantages. For example, they are easily accessible and therefore cost-effective, biodegradable and have low density [12]. The ease of modification and desirable aspect ratio of dimensions can also be mentioned as advantages of natural fibers. However, the non-homogeneous nature of the fibers can result in different defects of composites and/or complications during manufacturing. Natural fibers are hydrophilic. Thus hydrogen bonds could form, which can result in agglomerates or uneven distribution in the non-polar polymer matrix [18,20,21]. As most of the polymer composites are manufactured at high temperatures [19], the thermal instability of fibers poses some limitations to this process. Natural fibers can also be damaged by different bacteria, microorganisms, etc. To sum up, the modification of natural fibers should be carried out to improve their moisture resistance, interfacial adhesion, and compatibility with the polymer matrix [22,23,24].

Silanization is commonly used as a cellulose-based filler material modification method, as silanes as coupling agents improve the adhesion between the two phases—filler and polymer matrix—in composite [14,21,25]. Adhesion is improved as one part of the coupling agent attaches to the filler’s surface, and the other interacts with polymer [16,17].

In the space industry (e.g., launch systems), liquified hydrogen H_2_ and oxygen O_2_ are used as fuel. The gases need to be pressurized and cooled for space industry applications, so they do not occupy large volumes [25]. Temperatures under 123 K are considered cryogenic temperatures. As liquified H_2_ is stored at 20 K and liquified O_2_ at 89 K, fuel tanks need to be equipped with suitable insulation. Rigid PU foams have been chosen as insulation for tanks due to low thermal conductivity, cell structure, low density, and other parameters [24,25]. Rigid PU foams can be applied using a spray-on process, which can be considered an advantage because fuel tanks with complex shapes are entirely covered with insulation in such cases.

The flaws of rigid PU foams, e.g., dimensional instability, degradation in UV light, low mechanical properties, and a high coefficient of thermal conductivity, applicable for cryogenic insulation, can be compensated by using different composites. For example, rigid PU foams and glass fiber composites have shown improvements of elasticity modulus and flexural strength [26]. PU composites with graphene have shown an increase in compressive strength [27]. The addition of 0.6% of nanocrystalline cellulose improved flexural strength by 37% at 110 K [27].

In this study, sawdust (SD), microcellulose (MC), and nanocellulose (NC), as well as their silanized forms (SiSD, SiMC, and SiNC, respectively), were used to reinforce rigid PU foam composites to improve the material’s properties. The use of the developed PU composites as a cryogenic insulation material was investigated.

## 2. Materials and Methods

### 2.1. Materials

Rigid PU foam formulation used in this study was developed previously in the framework of a Bio4Cryo research project and is presented in the paper by Uram et al. [28]. The author named the developed rigid PU foam formulation PP40, and the same name is also kept for this paper. The characteristics of the developed PP40 composites were analyzed and compared to the properties of neat PP40.

TOFA (trade name FOR2) with low content of rosin acids (<2%) and unsaponifiables (<2%) was ordered from Forchem Oyj (Rauma, Finland). TEOA 99.2% was purchased from Huntsman (Rotterdam, The Netherlands).

TOFA-based polyol was synthesized as described in our previous work by Kirpluks et al. [29]. Polyol NEO 380 was used as an aromatic chain extender; it is commercially produced from recycled PET by Neo Group (Klaipėda, Lithuania). High functional polyol (HF) was synthesized at the Cracow Technical University (Poland) from rapeseed oil as described by Uram et al. [28]. DEG was purchased from Chempur (City, Germany).

A combination of two blowing agents, a chemical blowing agent water (water in polyols and added water) and a physical blowing agent Solkane^®^ 365/227 (Solvay, Brussels, Belgium) was used. A flame retardant tris(2-chloroisopropyl) phosphate (TCPP) purchased from Albermarle (Louvain-la-Neuve, Belgium) was used in rigid PU foam formulation. Tertiary amine-based catalyst Polycat^®^ 5 was purchased from Air Products and Chemicals Inc. (Rotterdam, The Netherlands). In addition, catalyst tin butyl dilaurate from Sigma–Aldrich (Steinheim, Germany) was added to rigid PU foam formulation. Poly-(dimethyldiphenyl)-diisocyanate (pMDI) Desmodur 44V20 L with isocyanate group content 31.5% was purchased from Covestro (Leverkusen, Germany).

MC pellets were obtained from birch kraft cellulose using oxidation with the ammonium persulphate method, as described by Filipova et al. [10]. NC (diameter 20–60 nm, length few µm) was ordered from Cellulose Lab (Fredericton, Canada). SD was obtained by dusting off wood pellets used for horse bedding. SD was delivered by Horse Span (Riga, Latvia). As a silanization agent, Dynasylan^®^SIVO 121 was used, which was kindly provided by Evonik (Essen, Germany).

### 2.2. Methods

#### 2.2.1. Silanization of Fillers

Filler, silanization agent, and distilled water were added in a 1:10:250 *w/v/v* ratio in the beaker and stirred for 2 h at room temperature (RT). Then, the mixture was filtered and oven-dried for 2 h at 50 °C. After drying, the product was washed two times with ethanol and once with distilled water, and filtered and oven-dried for 24 h at 50 °C.

#### 2.2.2. Characterization of Fillers

Untreated and silanized fillers were analyzed by determining their moisture content using the heated scales, A&D MX-50, obtaining images with scanning electron microscopy (SEM) using Edax Tescan 5136 MM microscope (Tescan, Brno, Czech Republic) (the samples were gilded before analyzing), and performing Fourier transmission infrared spectroscopy (FTIR) analysis using a Thermo Scientific Nicolet iS50 spectrometer (Waltham, MA, USA). FTIR was carried out using absorbance at a resolution 4 cm^−1^ with 32 scans. Particle size distribution of SD was also carried out by sieving using Retsch 200 mm × 50 mm sieves (Retsch GmbH, Haan, Germany).

#### 2.2.3. Preparation of Filler–Polyol Mixture Dispersions

Fillers were dispersed into a mixture of polyols. The dispersion process consisted of two steps. At first, the necessary amount (including possible mass losses of materials) of filler and mixture of polyols was weighed into a beaker. At first, the mixture was mixed with high shear mixer at 10,000 rpm for 10 min (the mixing rate was reached gradually). The second step was homogenizing the dispersion with ultrasound sonotrode (power 120 W, capacity of 40%, and amplitude of 40%) for 20 min. During both dispersion steps, the beaker was put into an ice bath to control the temperature.

#### 2.2.4. Characterization of Filler–Polyol Mixture Dispersions

Images with stereomicroscope LEICA S9i (Leica Biosystems, Wetzlar, Germany) were obtained of dispersions. Lightning and focal length were individually adjusted for each dispersion.

#### 2.2.5. Calculation of Isocyanate Index

In order to calculate the isocyanate index, the NCO value of pMDI, which was used in rigid PU foam formulation, has to be determined. The NCO value was determined according to ISO 14896:2006(E).

As the isocyanate index 110 was used [28], the necessary amount of pMDI was calculated by the following Equation (1) [3]:(1)$$m_{polyisocyanate}=\frac{II\cdot \Sigma OH\cdot m_{polyol}}{NCO\%\cdot 1336}+4.67\cdot m_{H_{2}O}$$
where:

*ΣOH*—total OH value of all polyols, mg KOH/g

$m_{H_{2}O}$—total mass of water in polyols and added water, g

#### 2.2.6. Preparation of Rigid PU Foam and Composites

Rigid PU foam samples were obtained according to the formulation presented in Table 1. Small size samples (cup-tests) were characterized using the universal foam qualification system Foamat 285, which measures different foaming parameters as foam height, apparent density, shrinkage, as well as start time, gel time, and end time.

At first, cup-tests of rigid PU foam samples were carried out. The required amount of PU components was weighed in 1 L paper cup and vigorously mixed for 10 s. Then cup was placed under the Foamat 285 ultrasound sensor, which measured the change of the foam’s height *H* and rising velocity *H’* as a function of time. The sample was weighed, and after 24 h the apparent density and shrinkage were measured by Foamat 285. 

#### 2.2.7. Mechanical Test Methods at 293 K and 77 K (Cryogenic Temperature)

A static Materials Testing Machine Z010 TN (Zwick GmbH & Co, Ulm, Germany) was used for foam testing at 293 K.

Compression tests were performed according to EN ISO 844:2014. Cylindric samples (diameter, 20 mm; height, 22 mm) were used for compression tests; tests were performed in two directions—parallel (Z) and perpendicular (X) to foam rise. Six samples in both directions were used (12 samples in total). Characteristics were determined using a preload of 1 kN and compressive rate of 10%/min.

Dumbbell type samples (length, 153 mm; thickness, 12 mm; width, 40 mm; and bottleneck, 26 mm) were used for tensile tests. In total, six samples were used, and tests were performed in a parallel direction.

A static Materials Testing Machine Z100 (Zwick GmbH & Co) was used for foam testing at 77 K.

For the compression test in 77 K, similar samples for testing at 293 K were used (see Figure 1 (left)). For tensile tests, ring-type samples were used (width, 13 mm; inner diameter, 43 mm; and outer diameter, 53 mm) as depicted in Figure 1 (right). Both tensile and compression samples were tested in an adjusted cryogenic facility, as shown in Figure 2. This test method has been described in detail by Uram et al. [28]. 

Obtained data from compression tests at both temperatures (modulus E and strength σ) were normalized with respect to density ρ of 40 kg/m^3^ [30]. The Equations (2) and (3) were used for the normalization of modulus (*E_norm_*) and strength (*σ_norm_*) [30]:(2)$$E_{norm}=E{\left(\frac{40}{\rho }\right)}^{1.7}$$
(3)$$\sigma_{norm}=\sigma {\left(\frac{40}{\rho }\right)}^{2.1}$$
where:

*E*—compressive modulus, MPa

*σ*—compressive strength, Mpa

*E_norm_*—normalized compressive modulus, Mpa

*σ_norm_*—normalized compressive strength, Mpa

*ρ*—density, kg/m^3^

#### 2.2.8. Adhesion Tests

Adhesion of the cryogenic insulation PU foam to aluminum plate was measured as tensile strength without cryo-shock and after cryo-shock (immersion in liquid nitrogen for 60 min), then rapidly warming to RT. Samples were tested according to EN 1607:2013. Foam material and aluminum plates had a total thickness of 20 mm. The sample was glued with PU adhesive between two plates (see Figure 3).

Samples were tested using a static Materials Testing Machine Z010 TN (Zwick GmbH & Co).

#### 2.2.9. Determination of Coefficient of Thermal Conductivity

The coefficient of thermal conductivity was determined using Linseis Heat flow meter 200 (Linseis, Selb, Germany), and a test was carried out according to ISO 8301:1991. The sample (dimensions 200 mm × 200 mm × 50 mm) was inserted between two plates (top plate temperature, 20 °C; bottom plate temperature, 0 °C). The coefficient of thermal conductivity was measured at +10 °C.

#### 2.2.10. Thermomechanical Analysis (TMA) and Safety Coefficient

TMA tests were performed using Linseis TMA PT (Linseis, Selb, Germany). Each sample was cut in a rectangular cuboid with h ~ 2 cm. The sample was cooled from 20 °C to −160 °C (3 °C/min) and then heated to 50 °C (3 °C/min). The shrinkage of material cooling it from 300 to 77 K is calculated from TMA results.

Safety coefficient is a characteristic that describes a material’s ability to maintain adhesion to aluminum after cryo-shock [25]. It is calculated by the following Equation (4) [25]:(4)$$k_{S}=\frac{\varepsilon_{77}}{\Delta l_{300-77}}$$
where:

*ε_77_*—tensile elongation at break at 77 K,%

Δ*l*_300-77_—shrinkage of material cooling it from 300 to 77 K,%

#### 2.2.11. Morphology of Foams

The morphology of foams (cell size) was analyzed using an optical microscope (OM) (PZO, Warsaw, Poland). The anisotropy index was calculated as the ratio of the cell heights and widths.

## 3. Results and Discussion

### 3.1. Fillers

At first, fillers used to obtain PP40 composites were characterized to determine different dissimilarities between untreated and silanized fillers. The obtained data helped to acknowledge how untreated and silanized fillers dispersed in polyol mixture and further affected rigid PU foam composites. 

#### 3.1.1. Particle Size Distribution of SD

The particle size distribution of SD was performed, and results are presented in Figure 4.

As it is depicted in Figure 4, over 75% of particles are 0–0.016 mm large. Therefore, it can be concluded that SD is mostly micro-sized filler.

#### 3.1.2. SEM

Images of untreated and silanized SD, NC, and MC are depicted in Figure 5, Figure 6 and Figure 7, respectively.

The surface of untreated SD particles was smoother (see Figure 5), but the surface of silanized particles was more rugged. The adhesion between the filler and polyol mixture can be improved by a rugged surface. Thus it is possible to obtain more homogenous dispersions. In the case of NC (see Figure 6), the fibres after silanization appeared to be thinner and shorter, which also can improve adhesion. After silanization, NC did not make agglomerates after being dispersed in polyols, deducing that their ability of water sorption has also decreased. Similar changes were also observed in the case of MC (see Figure 7).

#### 3.1.3. FTIR

Analysis with FTIR was used as the second method to prove that silanization has occurred. FTIR spectra of untreated and silanized fillers are depicted in Figure 8. The absorbance maximum around 800 cm^−1^ indicated –Si–CH_2_– bond vibration, which proved that silanization happened. This maximum is visible in spectra of SiSD and SiNC (see Figure 8 (left and middle, respectively)) but slightly less visible in spectra of SiMC (see Figure 8 (right)). Other absorbance maximums were typical of different bonds in cellulose. 

### 3.2. Filler–Polyol Mixture Dispersions

As mentioned before, fillers were dispersed into the mixture of polyols using a high shear mixer and ultrasonic sonotrode, and at first, composites with 1.5% filler concentration were obtained. Images of these filler–polyol mixture dispersions are depicted in Figure 9.

As it is depicted in Figure 9, dispersed NC and MC produced agglomerates. Therefore if rigid PU foams were going to be obtained with these fillers, composites would have significant defects. More homogenous dispersions were obtained with silanized fillers—SiSD, SiNC, and SiMC. Thus, these fillers were used for further trials. Dispersed SD stratified quite rapidly, which was why no further trials were carried out with this filler. 

In Figure 10, dispersions of other filler concentrations (for composites with 0.5–1.25% filler concentrations) are depicted. Physical-mechanical characteristics of the composites with 1.5% SiMC concentration were not satisfying. Therefore, it was not used for further trials.

As depicted in Figure 10, SiSD and SiNC dispersions in polyol mixture did not make either agglomerates stratified, and particle arrangement was rather even. Possible compactions may be decreased by optimizing the dispersion method.

### 3.3. Characteristics of Obtained Filler–PU Foam Composites

#### 3.3.1. Analysis of PU Composites with 1.5% Filler Concentration

As mentioned before, at first small size samples (cup tests) were obtained. Shrinkage and apparent density are presented in Table 2.

From the data presented in Table 2, it can be concluded that the addition of fillers increased both apparent density and shrinkage. Shrinkage was below 5%, which can be considered as a satisfying result. Also, the density of PP40 composites with fillers had increased, mainly because the viscosity of filler–polyol mixture dispersions increased. 

Bigger size (open mold) samples of PP40 composites with 1.5% SiSD, SiMC, and SiNC concentration were obtained. Physical-mechanical characteristics using test methods described previously were determined. In Table 2, the coefficient of thermal conductivity values are presented. PP40 composites had slightly lower values compared to neat PP40, though no significant improvement was visible. The decreased size of pores in composites may be considered as one of the reasons why the coefficient of thermal conductivity has reduced. The lower coefficient of thermal conductivity, the better composite insulates heat transmission. 

The compressive modulus and strength of obtained PP40 composites with 1.5% filler concentration are depicted in Figure 11.

As depicted in Figure 11 (left), the compressive modulus of PP40 composites in a perpendicular direction at RT was slightly higher than for neat PP40, but other properties were lower than for neat PP40 at RT. Compressive strength (see Figure 12 (right)) of PP40 composites with 1.5% filler concentration was lower than for neat PP40. At 77 K, the stiffness of foams increased. Thus both compressive modulus and strength were higher.

Tensile (Young’s) modulus and strength of neat PP40 and PP40 composites with 1.5% filler concentration at both RT and 77 K are depicted in Figure 12.

As shown in data depicted in Figure 12 (left), Young’s modulus of PP40 composites was higher than for neat PP40 at RT, but it was lower at 77 K. The tensile strength of PP40 composites was lower than for neat PP40 at both temperatures.

After examining the physical-mechanical properties of PP40 composites with 1.5% filler content, it was concluded that PP40 composites with SiSD and SiNC are more applicable for cryogenic insulation. Thus further experiments were carried out with these fillers.

#### 3.3.2. Analysis of PP40 Composites with 0.5–1.25% Filler Concentration

At first, the small size samples (cup-tests) of composites with 0.5–1.25% filler concentration were performed to measure density and shrinkage. These properties and the coefficient of thermal conductivity and safety coefficient (including data for calculation of safety coefficient) are presented in Table 3.

As can be determined from data presented in Table 3, thermal conductivity of few composites, e.g., PP40 composite with 1.5% SiSD concentration, has improved. Over all, most of the composites show no more than ~5% improvement of the coefficient of thermal conductivity. This can be explained by the different morphology of developed rigid PU foam. The pore size of PP40 composites has decreased (see Section 3.3.3), with the addition of fillers.

The compressive modulus of PP40 composites with 0.5–1.25% filler concentration at RT and 77 K are presented in Figure 13.

There was a slight increase of both compressive modulus and strength in the case of PP40 composites with SiNC. However, this change was observed only for composites with SiNC till 1.25% concentration of filler. In the case of PP40 composites with SiSD, a similar improvement was also observed, though less expressed. However, PP40 composite with 0.75% SiSD concentration had the highest compressive modulus in parallel direction at 77 K. 

Similar to previous results of compressive modulus, a slight improvement of compressive strength was also noticed in the case of PP40 composites with SiNC until 1.25% concentration of filler (see Figure 14) at RT. However, overall compressive strength at 77 K did not give an acknowledgeable improvement of PP40 composite characteristics.

Tensile (Young’s) modulus and strength of composites with 0.5–1.25% concentration and PP40 are depicted in Figure 15. Slight improvement of Young’s modulus can be seen in the case of SiSD at RT (see Figure 15). Overall, Young’s modulus and tensile strength of PP40 composites was lower than for neat PP40.

Some of the individual properties of PP40 composites have been improved by adding fillers, e.g., the safety coefficient of PP40 composite with 1.25% SiNC concentration was higher than for neat PP40. However, overall, no significant improvement of properties has been observed. The safety coefficient is characteristic of the material, which described its ability to maintain adhesion to aluminum plate after cryo-shock. Therefore, it is believed that this composite is more applicable for cryogenic insulation, and the adhesion tests were performed for this composite. The strength of adhesion is depicted in Figure 16.

#### 3.3.3. Morphology of PU Composites

Morphology of PU composites was carried out by OM. The anisotropy index of the neat PP40 and PP40 composites is presented in Table 4. 

The addition of fillers had no significant impact on the anisotropy index in both parallel (Z) and perpendicular (X) directions.

The images of PP40 obtained with an OM are depicted in Figure 17.

Cells in the Z direction were more evenly distributed. The anisotropy index of neat PP40 in the Z direction was higher, which correlates with increased mechanical characteristics in the Z direction (compressive modulus and strength). In the X direction, few imperfections were observed. The images of neat PP40 and SiSD composites obtained with OM are depicted in Figure 18.

In all PP40 composites with SiSD, few defects were observed. Similarly to neat PP40, PP40 composites with SiSD presented slightly higher mechanical properties in the Z direction, which can be associated with an increased anisotropy index (see Table 4). The images of PP40 composites with SiMC and SiNC obtained with OM are depicted in Figure 19.

In all PP40 composites with SiMC and SiNC, few defects were observed. Similarly to neat PP40, PP40 composites with SiNC presented slightly higher mechanical properties in the Z direction, which can be associated with an increased anisotropy index (see Table 4).

After analyzing the obtained results, it can be concluded that physical and mechanical properties of the PP40 composites overall, have not improved. This can be explained by the distortions in the cell structure in PP40 and filler composites. The addition of the filler in the rigid PU foam significantly increased the viscosity of the mixture. This hinders the formation of even size cells. Furthermore, the filler introduces local defects of the rigid PU foam structure, which negatively effects the mechanical properties of the PP40 composite.

## 4. Conclusions

Three different fillers—SD, MC, and NC—and their silanized forms were dispersed into a polyol mixture using a high shear mixer and ultrasound sonotrode, and rigid PU foam composites were obtained. It was observed that silanized fillers produced no agglomerates. Therefore more homogenous dispersions were obtained.

Comparing composites obtained with untreated fillers and composites obtained with silanized fillers, it was observed that composites reinforced with silanized fillers produced no significant defects, owing to the more homogenous dispersions of silanized fillers and polyol mixture. PP40 composites were reinforced with 0.5–1.5% concentration of fillers. After the first attempts, when PP40 composites with 1.5% concentration of fillers were developed, it was noticed that PP40 composites strengthened with SiSD and SiNC showed better characteristics than PP40 composites with SiMC or untreated fillers. Therefore, further trials obtaining PP40 composites with 0.5–1.25% concentration of fillers were carried out using SiSD and SiNC. 

After testing physical mechanical characteristics of composites, it was found that the lowest apparent density was observed when 1.5% of SiMC was introduced in rigid PP40 foam (51.3 kg/m^3^), and the highest safety coefficient was observed in composite with 1.25% SiNC concentration (3.14). Unfortunately, no significant improvement of physical mechanical characteristics of PP40 composites with fillers was observed. Further trials of the cellulose-based filler impact on PU composites’ characteristics and possible solutions of unsatisfactory results should be investigated. 

## Figures and Tables

**Figure 1 polymers-13-04089-f001:**
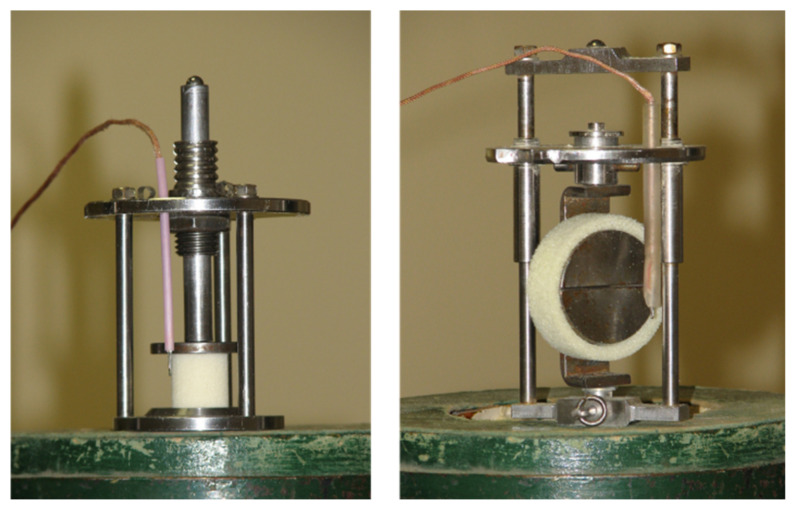
Appliance for samples for compression (**left**) and tensile (**right**) tests in 77 K.

**Figure 2 polymers-13-04089-f002:**
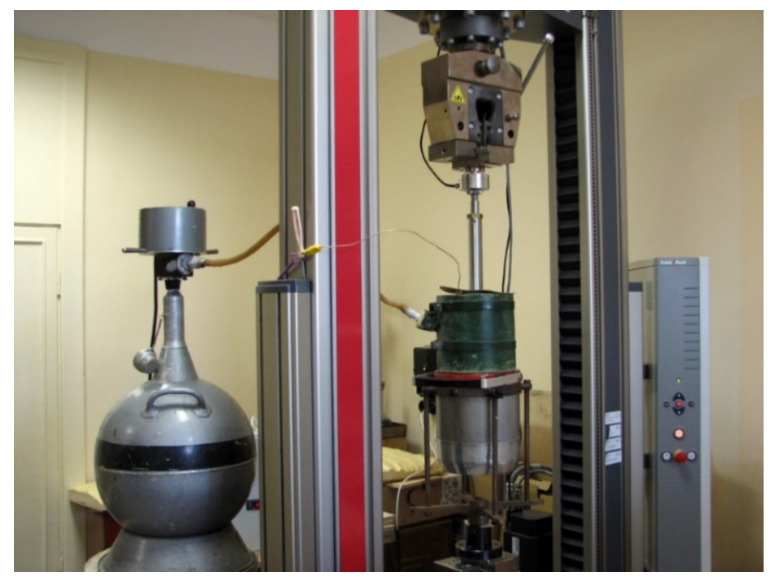
Testing machine equipped with cryogenic insulation facility for tests at 77 K.

**Figure 3 polymers-13-04089-f003:**
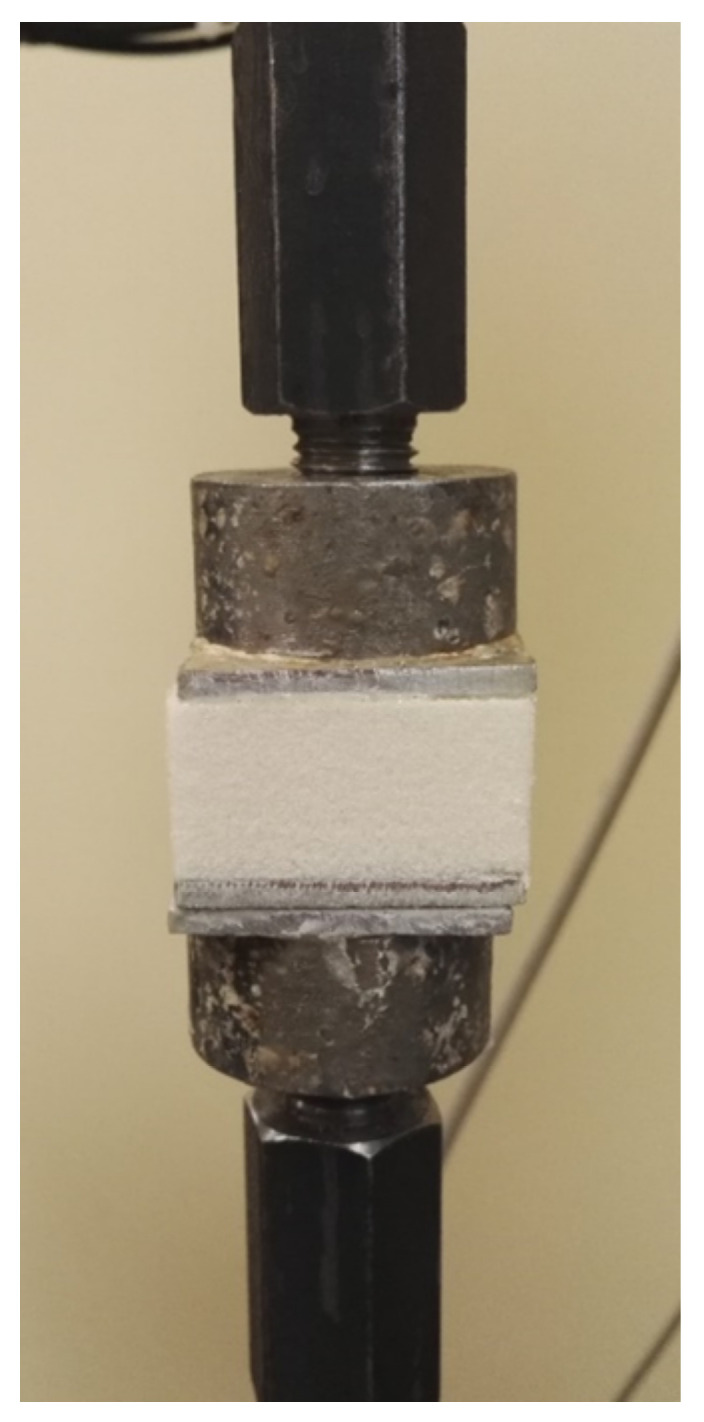
Cryogenic insulation rigid PU foam sample prepared for adhesion test.

**Figure 4 polymers-13-04089-f004:**
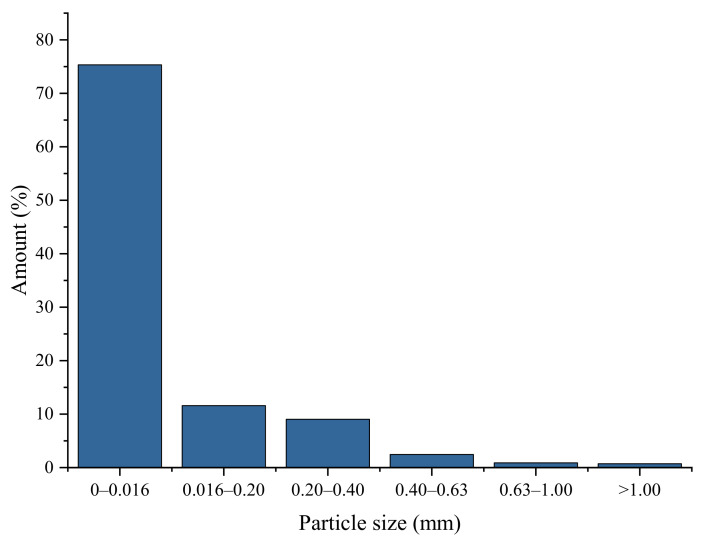
Particle size distribution of sawdust (SD).

**Figure 5 polymers-13-04089-f005:**
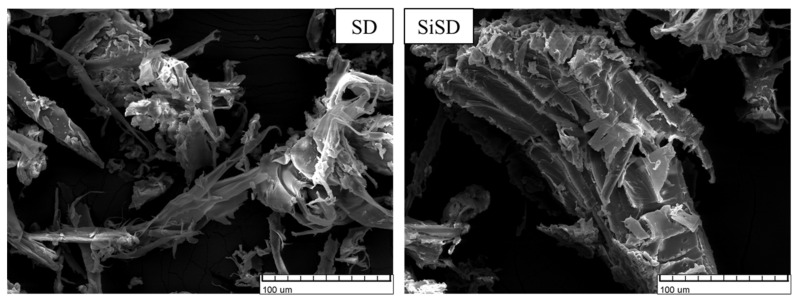
SEM images of sawdust (SD) and silanized sawdust (SiSD).

**Figure 6 polymers-13-04089-f006:**
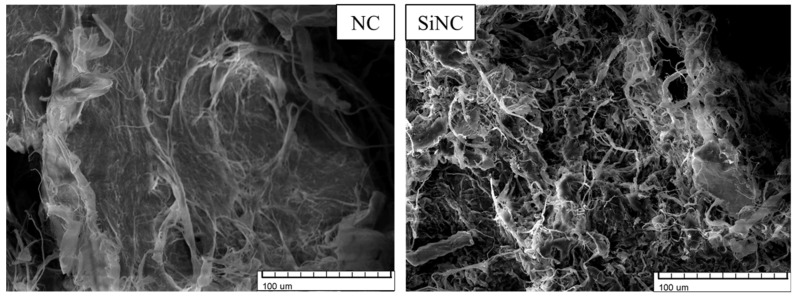
SEM images of nanocellulose (NC) and silanized nanocellulose (SiNC).

**Figure 7 polymers-13-04089-f007:**
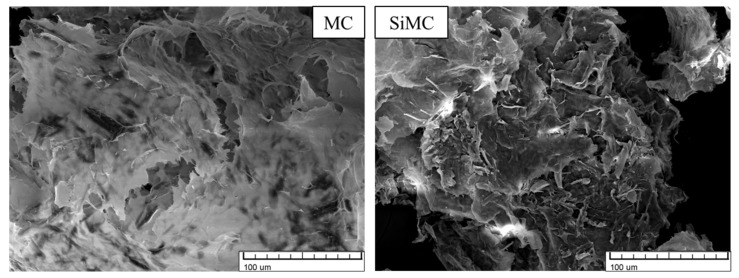
SEM images of microcellulose (MC) and silanized microcellulose (SiMC).

**Figure 8 polymers-13-04089-f008:**
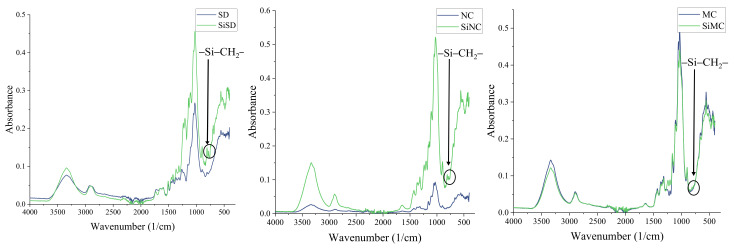
FTIR spectra of SD and SiSD (**left**), NC and SiNC (**middle**), and MC and SiMC (**right**).

**Figure 9 polymers-13-04089-f009:**
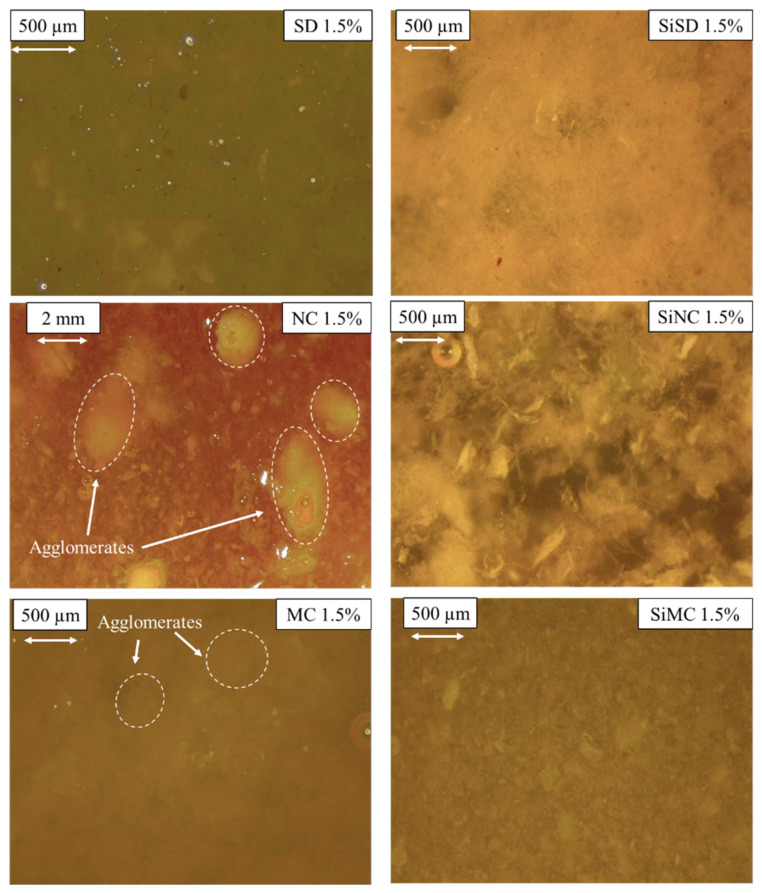
Dispersions of fillers in polyol mixture for 1.5% PP40 composites.

**Figure 10 polymers-13-04089-f010:**
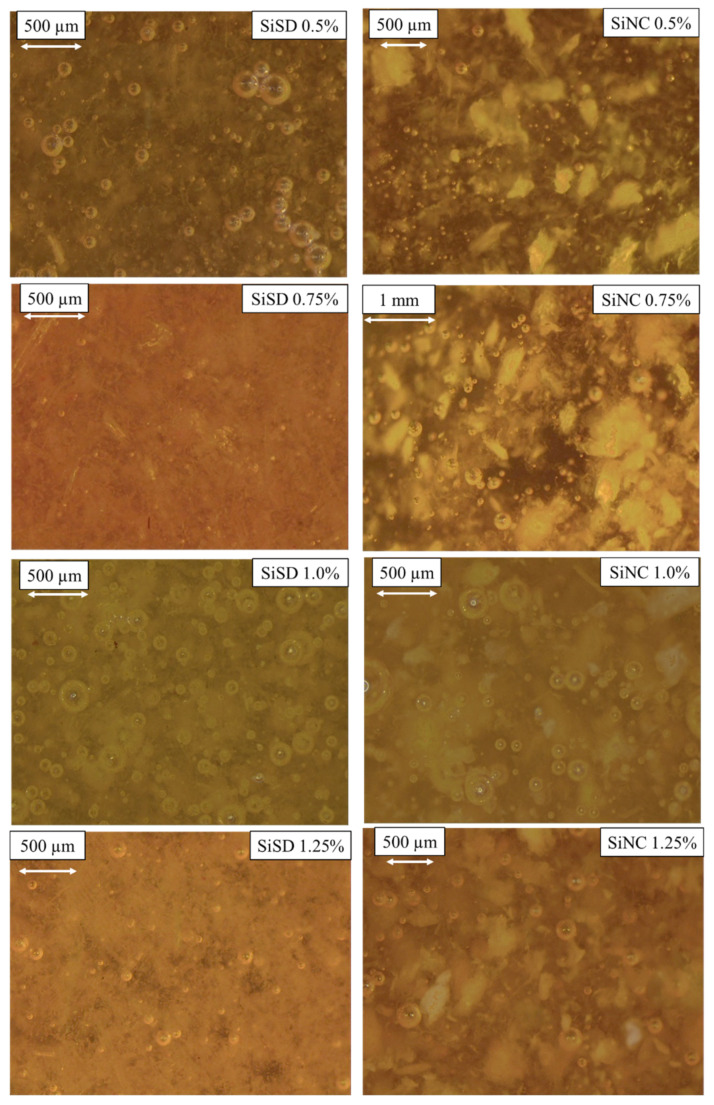
Dispersions of fillers in polyol mixture for 0.5%, 0.75%, 1.0%, and 1.25% PP40 composites.

**Figure 11 polymers-13-04089-f011:**
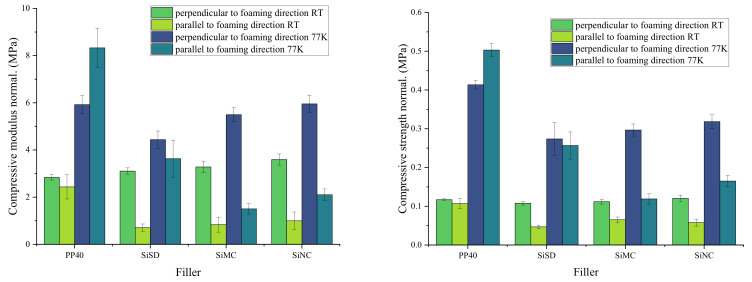
Compressive modulus (**left**) and strength (**right**) of neat PP40 and PP40 composites with 1.5% filler concentration.

**Figure 12 polymers-13-04089-f012:**
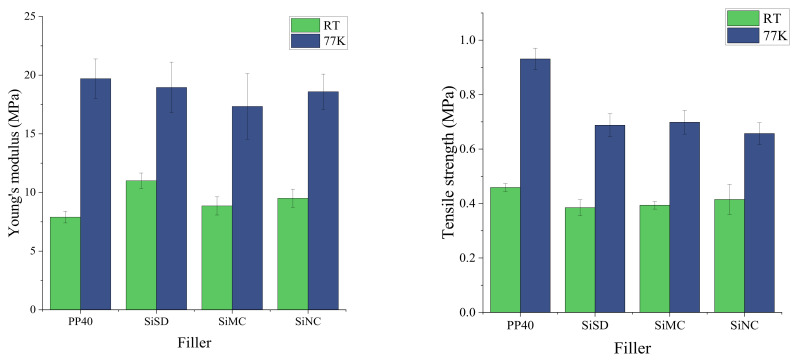
Young’s modulus (**left**) and tensile strength (**right**) of neat PP40 and PP40 composites with 1.5% filler concentration.

**Figure 13 polymers-13-04089-f013:**
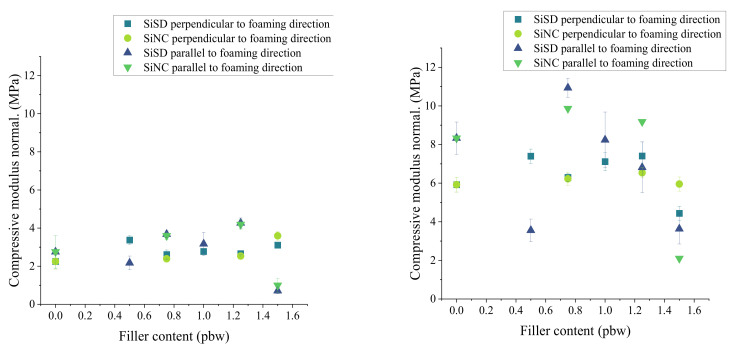
The compressive modulus of PP40 composites with 0.5–1.25% filler concentration at RT (**left**) and 77 K (**right**).

**Figure 14 polymers-13-04089-f014:**
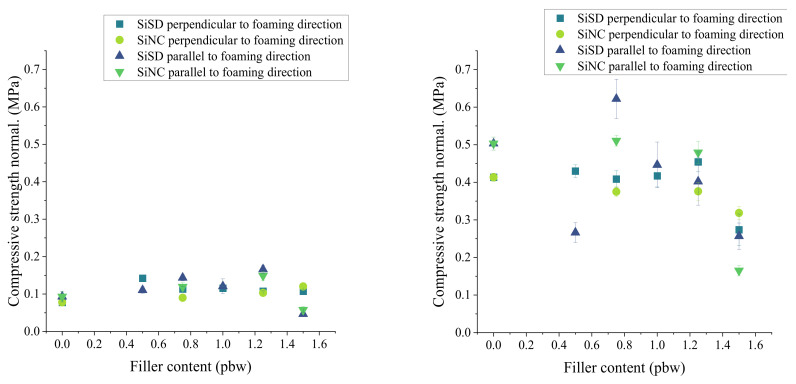
The compressive strength of PP40 composites with 0.5–1.25% filler concentration at RT (**left**) and 77 K (**right**).

**Figure 15 polymers-13-04089-f015:**
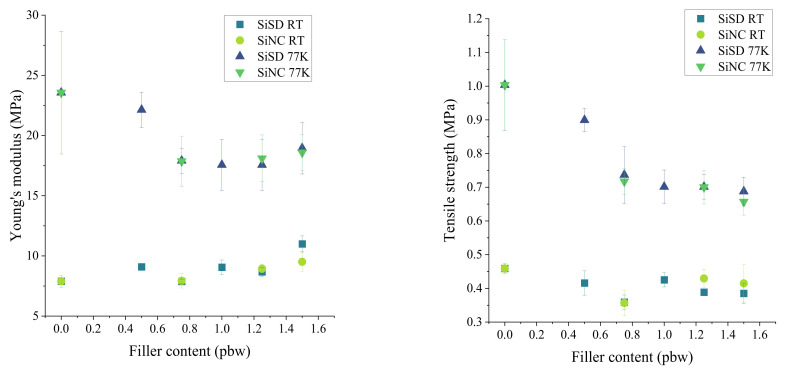
Tensile (Young’s) modulus (**left**) and strength (**right**) of PP40 composites with 0.5–1.25% filler concentration at RT and 77 K.

**Figure 16 polymers-13-04089-f016:**
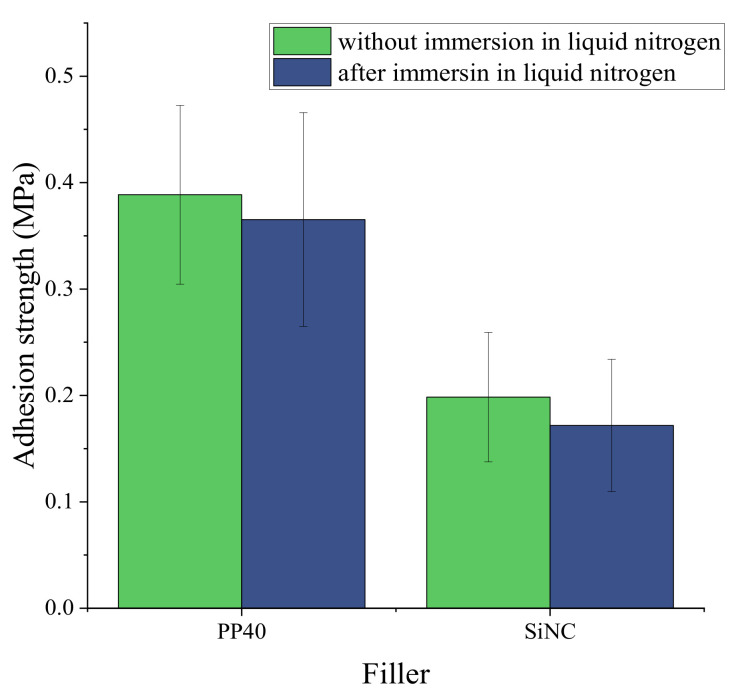
The adhesion strength of neat PP40 and PP40 composite with 1.25% silanized nanocellulose (SiNC) concentration.

**Figure 17 polymers-13-04089-f017:**
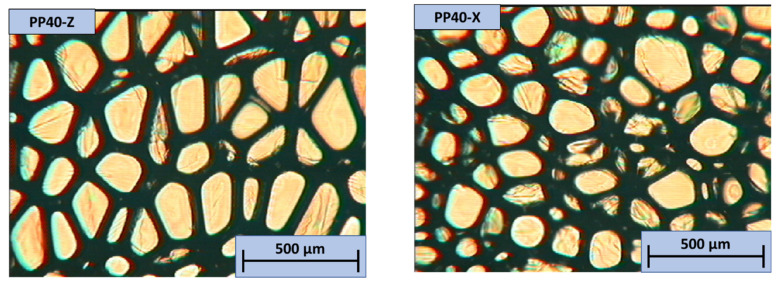
OM images of neat PP40 in both Z (**left**) and X (**right**) directions.

**Figure 18 polymers-13-04089-f018:**
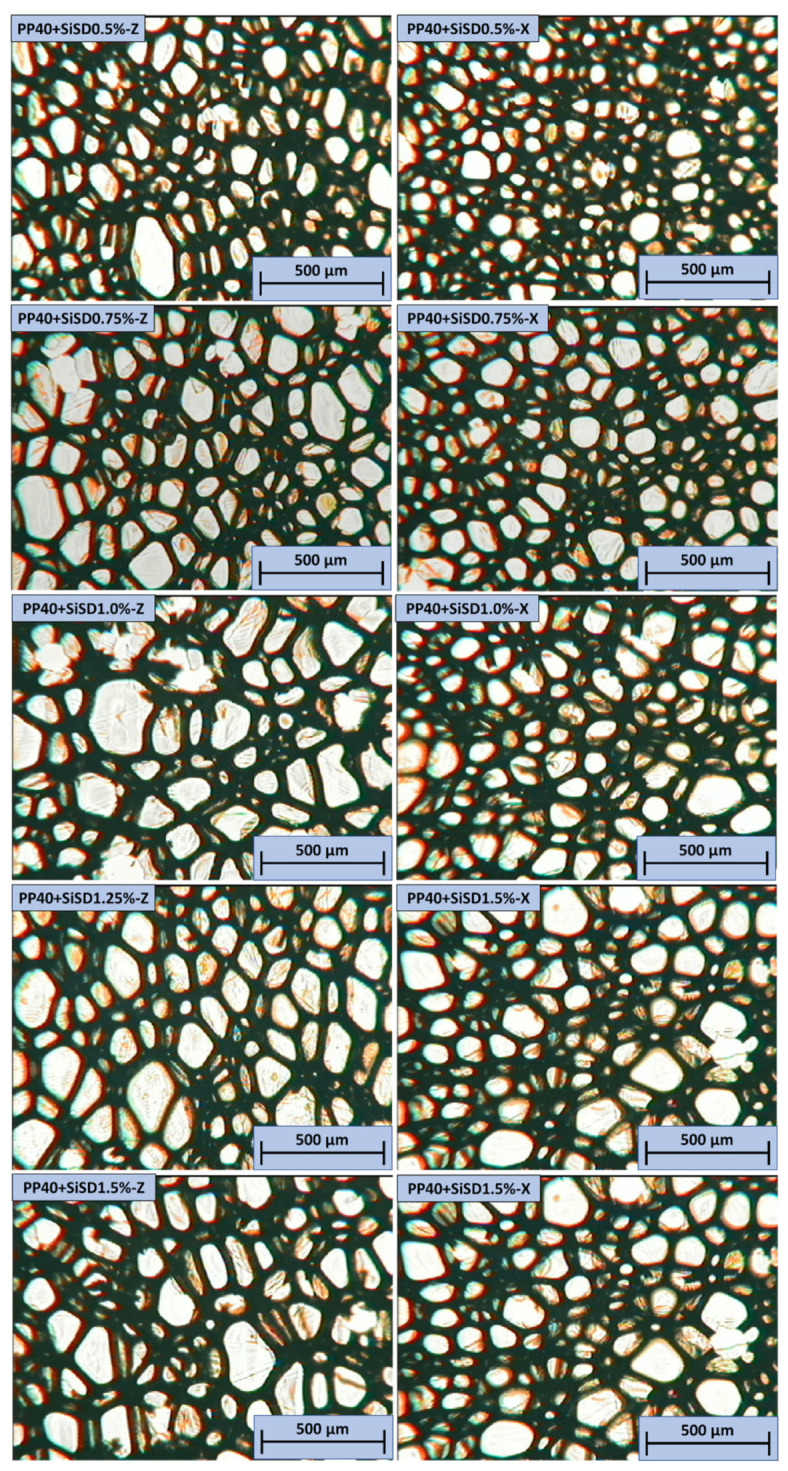
OM images of neat PP40 and SiSD composites in both Z (**left**) and X (**right**) directions.

**Figure 19 polymers-13-04089-f019:**
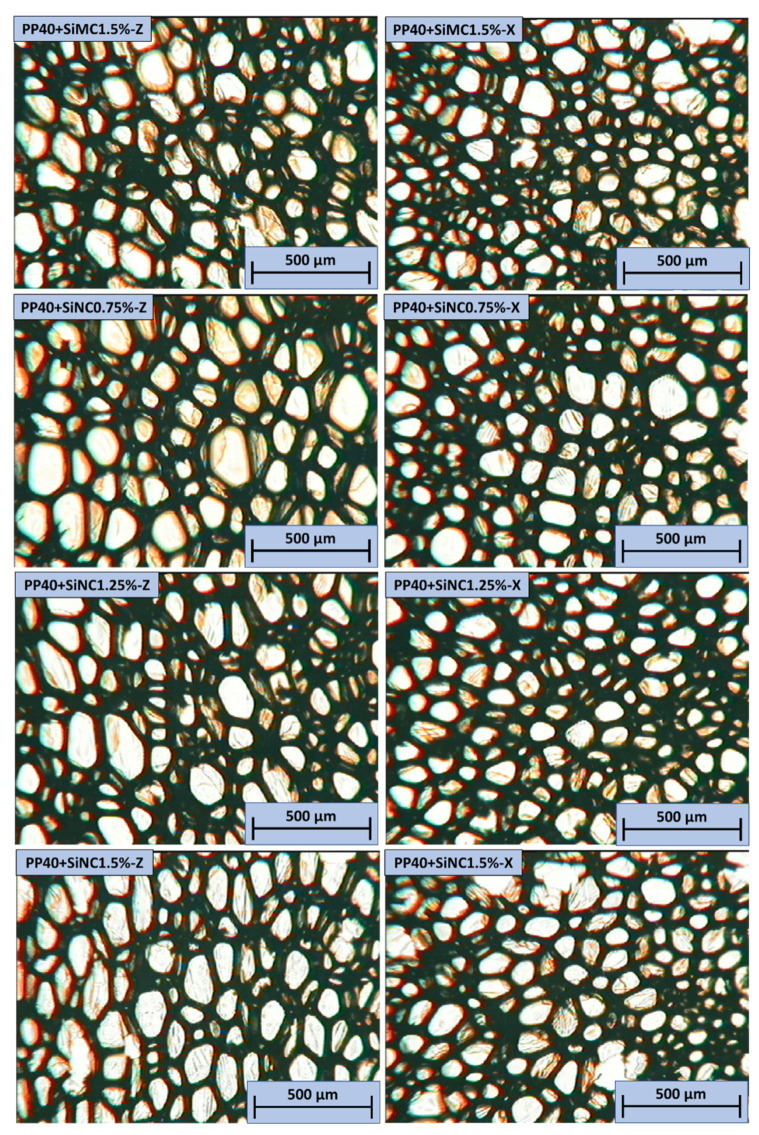
OM images of neat PP40 and SiMC and SiNC composites in both Z (**left**) and X (**right**) directions.

**Table 1 polymers-13-04089-t001:** Rigid PU formulation PP40 [28].

Components		Amount, pbw	Main Characteristics
Polyols	ETOFA/TEOA	25	Renewable material content 60.7%, OH value 449 mg KOH/g, water content 0.14%
	HF	30	Renewable material content 69.1%, OH value 267 mg KOH/g, water content 0.40%
	DEG	25	OH value 1057 mg KOH/g, water content 0.17%
	NEO 380	20	Recycled material content 50%, OH value 366 mg KOH/g, water content 0.21%
Flame retardant	TCPP	15	CAS: 13674-87-8
Blowing agents	Solkane^®^ 365/227	25	Blend of 1,1,1,3,3-pentafluorobutane and 1,1,1,2,3,3,3-heptafluoropropane
	Water	0.5	-
Catalysts	Polycat^®^ 5	0.5	CAS: 3030-47-5
	Tin butyl dilaurate	0.01	CAS: 77-58-7
Surfactant	L-6915	1.5	Organosilicon surfactant
Isocyanate index		110	-
Polyisocyanate	pMDI	147	Isocyanate group content 31.5%
Fillers	SD, SiSD, NC, SiNC, MC, SiMC	0.5–1.5 wt%	SD, SiSD—brown, solid particles NC, SiNC, MC, SiMC—white, solid particles

**Table 2 polymers-13-04089-t002:** Apparent density, shrinkage, and coefficient of thermal conductivity of PP40 composites with 1.5% filler concentration and neat PP40.

Composite	Apparent Density ρ, kg/m^3^	Shrinkage,%	Coefficient of Thermal Conductivity λ, mW/(m ∙ K)
Neat PP40	52.7	0.2	18.72
PP40 + SD1.5%	56.3	4.9	-
PP40 + SiSD1.5%	60.1	2.6	17.65
PP40 + MC1.5%	57.1	1.8	-
PP40 + SiMC1.5%	51.3	2.4	18.45
PP40 + NC1.5%	58.5	1.4	-
PP40 + SiNC1.5%	55.5	3.3	18.10

**Table 3 polymers-13-04089-t003:** Coefficient of thermal conductivity, density, tensile elongation at break at 77 K, shrinkage of material cooling it from 300 to 77 K, and safety coefficient of PP40 composites and neat PP40.

Composite	Coefficient of Thermal Conductivity λ, mW/(m∙K)	Density ρ, kg/m^3^	Tensile Elongation at Break at 77 K ε_77_,%	Shrinkage of Material Cooling it from 300 to 77 K Δl_300-77_, %	Safety Coefficient k_s_
Neat PP40	18.72	52.7	5.1	1.86	2.73
PP40 + SiSD0.5%	18.43	61.4	3.5	1.65	2.11
PP40 + SiSD0.75%	18.59	58.3	3.7	1.62	2.28
PP40 + SiSD1.0%	18.89	55.5	4.1	1.48	2.78
PP40 + SiSD1.25%	18.69	58.2	3.9	1.35	2.86
PP40 + SiSD1.5%	17.65	60.1	3.3	1.49	2.25
PP40 + SiMC1.5%	18.45	51.3	3.1	1.83	1.72
PP40 + SiNC0.75%	18.16	58.4	4.0	1.69	2.36
PP40 + SiNC1.25%	18.93	54.7	3.9	1.24	3.14
PP40 + SiNC1.5%	18.10	55.5	3.5	1.62	2.18

**Table 4 polymers-13-04089-t004:** Anisotropy index of neat PP40 and PP40 composites.

Composite	Z	X
	Anisotropy Index	Anisotropy Index
Neat PP40	1.22 ± 0.0392	0.94 ± 0.034
PP40 + SiSD0.5%	1.22 ± 0.0260	0.85 ± 0.082
PP40 + SiSD0.75%	1.16 ± 0.0216	0.90 ± 0.021
PP40 + SiSD1.0%	1.03 ± 0.0830	0.87 ± 0.021
PP40 + SiSD1.25%	1.27 ± 0.0357	0.88 ± 0.018
PP40 + SiSD1.5%	1.15 ± 0.0632	0.85 ± 0.034
PP40 + SiMC1.5%	1.15 ± 0.125	0.90 ± 0.023
PP40 + SiNC0.75%	1.23 ± 0.102	0.99 ± 0.0085
PP40 + SiNC1.25%	1.33 ± 0.0987	0.83 ± 0.011
PP40 + SiNC1.5%	1.42 ± 0.139	0.85 ± 0.014

## Data Availability

The data presented in this study are available on request from the corresponding author.

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
