# Peer review of "Polyurethane Foam Composites Reinforced with Renewable Fillers for Cryogenic Insulation"

_polymers, 2021, doi:10.3390/polym13234089_

Round 1
Reviewer 1 Report
In this paper, sawdust, microcellulose and nanocellulose and their silanized forms were used to reinforce rigid polyurethane (PU) foam composites. For rigid PU foam formulations, three polyols from recycled and renewable materials were used. The physical-mechanical properties such as elasticity modulus, compressive and tensile strength in both 293 K and 77 K, adhesion measurements with and without cryo-shock, apparent density, thermal conductivity coefficient, and safety coefficient were measured. The manuscript could be considered for publication based on the novelty and truth of its content after minor revision.
My detailed comments are as follows:
(1)Whether the heat insulation of PP40 composites was reinforced by SISD and SINC improve significantly? •Please give corresponding explanation.
(2)Why the physical and mechanical properties of PP40 composites with fillers have not been improved? The corresponding supplementary explanations should be given in this section.
Author Response
Author's response to the reviewer comments are the following:
(1)
As it can be determined form data presented in Table 3.2, thermal conductivity of few composites, e.g., PP40 composite with 1.5 % SiSD concentration, has improved. Over all, most of the composites show no more than ~5 % improvement of the coefficient of thermal conductivity. This can be explained by the different morphology of developed rigid PU foam. The pore size of PP40 composites has decreased (see 3.3.3 paragraph), with addition of fillers
(2)
After analyzing the obtained results, it can be concluded that physical and mechanical properties of PP40 composites over all have not improved. This can be explained by the distortions in the cell structure in PP40 and filler composites. Addition of the filler in the rigid PU foam significantly increased the viscosity of the mixture. This hinders the formation of even size cells. Furthermore, the filler introduces local defects of the rigid PU foam structure, which negatively effects the mechanical properties of the PP40 composite.
Both of the answers have also been added to the manuscript, which is attached to this message.

Reviewer 2 Report
The paper entitled “Polyurethane Foam Composites Reinforced With Renewable Fillers for Cryogenic Insulation”. The paper is well organized and written. Obtained results are well presented and described. In my opinion, this work could be published in the present form. Below you can find some comments which can improve this work.
Results and discussion:
- Materials and methodology: testing methods for filler and filler-polyol mixture dispersions characterization should be presented separately (in the same style as for foams)
- References: In my opinion references include too many self-citation.
Author Response
The requested changes in the manuscript have been made.
Please see the attachment.
